# Differing Escape Responses of the Marine Bacterium *Marinobacter adhaerens* in the Presence of Planktonic vs. Surface-Associated Protist Grazers

**DOI:** 10.3390/ijms231710082

**Published:** 2022-09-03

**Authors:** Luis Alberto Villalba, Minoru Kasada, Luca Zoccarato, Sabine Wollrab, Hans Peter Grossart

**Affiliations:** 1Department Plankton and Microbial Ecology, Leibniz Institute of Freshwater Ecology and Inland Fisheries (IGB), 16775 Stechlin, Germany; 2Graduate School of Life Sciences, Tohoku University, 6-3, Aoba, Sendai 980-8578, Japan; 3Institute of Computational Biology, University of Natural Resources and Life Sciences (BOKU), Muthgasse 18, 1190 Vienna, Austria; 4Institute for Biochemistry and Biology, Potsdam University, 14482 Potsdam, Germany

**Keywords:** pelagic environment, microbial loop, bacterial lifestyles, adaptive dynamics, inducible defense, habitat choice, predator-prey interactions, phenotypic plasticity, bacterial defensive mechanisms

## Abstract

Protist grazing pressure plays a major role in controlling aquatic bacterial populations, affecting energy flow through the microbial loop and biogeochemical cycles. Predator-escape mechanisms might play a crucial role in energy flow through the microbial loop, but are yet understudied. For example, some bacteria can use planktonic as well as surface-associated habitats, providing a potential escape mechanism to habitat-specific grazers. We investigated the escape response of the marine bacterium *Marinobacter adhaerens* in the presence of either planktonic (nanoflagellate: *Cafeteria roenbergensis*) or surface-associated (amoeba: *Vannella anglica*) protist predators, following population dynamics over time. In the presence of *V. anglica*, *M. adhaerens* cell density increased in the water, but decreased on solid surfaces, indicating an escape response towards the planktonic habitat. In contrast, the planktonic predator *C. roenbergensis* induced bacterial escape to the surface habitat. While *C. roenbergensis* cell numbers dropped substantially after a sharp initial increase, *V. anglica* exhibited a slow, but constant growth throughout the entire experiment. In the presence of *C. roenbergensis*, *M. adhaerens* rapidly formed cell clumps in the water habitat, which likely prevented consumption of the planktonic *M. adhaerens* by the flagellate, resulting in a strong decline in the predator population. Our results indicate an active escape of *M. adhaerens* via phenotypic plasticity (i.e., behavioral and morphological changes) against predator ingestion. This study highlights the potentially important role of behavioral escape mechanisms for community composition and energy flow in pelagic environments, especially with globally rising particle loads in aquatic systems through human activities and extreme weather events.

## 1. Introduction

In bacterial communities, species that can rapidly escape predation by phagotrophic protists, for example, via phenotypic plasticity, have a competitive advantage under high predation pressure. Protist grazing effectively controls bacterial biomass (e.g., [1]) and enables the transport of a substantial part of the bacterial biomass to higher trophic levels through the “microbial loop” [2]. Therefore, bacteria–protist trophic interactions have profound consequences for organic matter and energy flow, as well as food-web structure in aquatic systems [3]. Furthermore, in natural systems, it has been shown that predator-driven escape mechanisms are a key driver of shaping bacterial communities in aquatic environments (e.g., [4]). For example, species with high morphological plasticity have the ability to reduce grazing-induced mortality compared to species with low plasticity [4], which in turn modifies bacterial community composition under high predation events. Some bacterial species have evolved rapid defense mechanisms in order to avoid grazing pressure, such as antibiotic resistance, changes in cell size and motility patterns, cell clumping, and exopolymer formation [4,5,6]. In the pelagic environment, protist predators are often specialized in grazing on free-floating planktonic bacteria or on surface-associated bacteria, e.g., on marine particles. For bacteria with the ability to use both habitats [7], this offers the possibility of a behavioral escape mechanism in response to high predation pressure in one habitat [8]. The escape behavior might thereby be triggered by the recognition of protist predators via chemical cues [9].

In aquatic pelagic environments, carbon and nutrient substrates are present in liquid and solid states—defined as dissolved organic matter (DOM) and particulate organic matter (POM)—and their densities vary over space and time [10,11]. To exploit these two pools of organic matter, bacteria have developed different life strategies, either specialized in a specific habitat or the ability to exploit both habitats. In this study, specialist and generalist bacteria are discriminated according to their habitat choice (i.e., free-living (planktonic) vs. surface-associated) and not to their substrate choice, as has been previously done. According to our definition, specialist bacteria are either specialized in a planktonic (e.g., [12]) or surface-associated [13] lifestyle (i.e., utilizing either dissolved or particulate organic matter). In contrast, the generalist strain can be planktonic and surface-associated at the same time, and thus able to move between the two different compartments, e.g., to avoid grazing pressure by planktonic or surface-associated grazers. Consequently, generalist bacteria are physiologically adapted to utilize dissolved and particulate organic matter at the same [7]. Thereafter, specialists are more effective in exploiting a special type of resource with the cost of a strong substrate-specific dependence. In contrast, generalists may be less efficient in exploitation of a specific resource (due to higher metabolic maintenance costs for critical structures/metabolic pathways), but can more effectively escape grazing pressure in one or the other habitat. However, the ability to shift between habitats in response to predation requires motility via flagella, as well as chemotaxis to find new suitable living-habitats. Furthermore, chemical cues might enable them to sense the presence of grazers, which could trigger a habitat shift [8]. Such a predator-escape mechanism could be an important advantage compared to habitat specialists, especially under high grazing pressure. Assessing and understanding bacterial defense mechanisms is important, due to their far-reaching ecological implications for the dynamics and composition of the microbial community [14], as well as for energy transport to higher trophic levels [15]. Thereby, microbial community composition and dynamics can be influenced by habitat heterogeneity determined by particle load and type. Worldwide, an increase in particle loading to aquatic systems can be observed due to anthropogenic impacts, such as microplastic pollution (e.g., [16]), or following extreme events, such as flooding, which occur at increasing frequency as part of climate change [17,18]. Increasing particle concentrations can be expected to increase bacterial abundance and correspondingly lead to an increase in their respective grazers. Theoretical studies suggest that this might benefit generalist bacterial species with the ability to escape from habitats with high predation pressure, with potential consequences also for energy transport through the microbial loop [15]. To assess the ecological consequences, it is important to perform observational studies on bacterial habitat choice and how it is influenced by predation. The difficulty of assessing bacterial behavioral dynamics in nature points to the necessity of expanding our knowledge by using laboratory or modeling approaches [19]. The impact of bacterial escape mechanisms on protist biomass can serve as a proxy for energy available for higher trophic levels.

Here, we used a widespread marine bacterium, *Marinobacter adhaerens*, which is known to dwell both on particle surfaces and in water, to experimentally investigate its rapid behavioral defense mechanisms in response to predation by a planktonic or a surface-associated phagotrophic protist, respectively. We used the flagellate *Cafeteria roenbergensis* as a model predator for the planktonic habitat, which primarily feeds on planktonic bacteria by creating feeding currents with its anterior flagellum [20,21]. For a surface-associated predator, we used the amoeba *Vannella anglica*, a surface-crawler predator that feeds on biofilm-forming bacteria [22]. Flexible habitat choice and the ability to thereby escape predation might be one reason for the widespread occurrence of *M. adhaerens* and other species of the same genus [23]. To assess the importance of this escape mechanism, we investigated the habitat use and population dynamics of *M. adhaerens* in the absence and presence of planktonic or surface-associated protist predators. We used nutrient-poor medium to mimic nutrient conditions typically found in marine environments and added nutrient-enriched organic particles. We observed different escape mechanisms expressed by *M. adhaerens* in the presence of planktonic vs. surface-associated protist predators. Findings from this study highlight the importance of habitat heterogeneity in pelagic environments, which can create an opportunity of behavioral escape mechanisms via flexible habitat choice for generalist bacteria. This has potential implications for microbial community composition and consequently for energy transfer through the microbial loop [15].

## 2. Results

Bacterial population dynamics in the presence of predators.

### 2.1. Planktonic Predator Treatment

In the control as well as the planktonic predator treatment, after the initial inoculation of *M. adhaerens*, its planktonic cell density increased during the first two days (Figure 1A). In both cases, after the second day of incubation, the cell density of *M. adhaerens* in the water column declined sharply until day 3, settling on a relatively constant level of low population density until the end of the experiment. The surface-attached *M. adhaerens* subpopulation (Figure 1B) grew steadily on the agar surfaces until day 6 in the planktonic predation treatment and until day 11 in the control treatment. Maximum bacterial densities attached to surfaces are substantially higher in the planktonic predation scenario. Thereafter, cell densities of control and predator treatment started to decline until the end of the experiment. Following addition of *C. roenbergensis* on day 2 (Figure 1C), predator cell density showed a sharp increase, reaching its maximum on day 4, after which *C. roenbergensis* cell density dropped sharply on day 5 and continued to drop to low densities during the rest of the experiment. Albeit considered a planktonic protist, *C. roenbergensis* was also feeding on bacteria associated with the water–surface interphase. It was observed that a subpopulation of *C. roenbergensis* stayed in the vicinity of the nutrient-rich agar surfaces (see Appendix A). This subpopulation showed a similar trend, reaching maximum density on day 5 (one day later compared to the rest of the population) and a decrease until day 10, after which the predator subpopulation close to the agar surface disappeared completely (see Appendix A).

Comparing the control and predation treatments shows that predator effects on bacterial density in the planktonic habitat were not pronounced. The GAM analysis shows no significant effect of *C. roenbergensis* on planktonic *M. adhaerens* (Figure 2A, df = 2.817, F = 1.263, *p* = 0.31), although cell density of *M. adhaerens* is lower in the presence of the planktonic predator compared to the control (Figure 1A). The strongest difference between control and planktonic predator treatment is observed for the surface habitat, where *M. adhaerens* density reaches much higher densities in the presence of the planktonic predator compared to the control (see Appendix A), indicating increased bacterial attachment to surfaces in the presence of the grazer (Figure 1B). This is supported by the GAM analysis, which shows a significant positive correlation of *C. roenbergensis* density with areal occupation of the surface-attached *M. adhaerens* (Figure 3A, df = 4.100, F = 11.145, *p* < 0.001). On day 6, the increase in bacterial density on the agar plate coincides with the maximum predator biomass in the water–surface interphase (see Appendix A). After day 9, when predator density was close to extinction, *M. adhaerens* density in the planktonic (Figure 1A) and surface-attached habitat (Figure 1B) was similar to observed densities in the control treatment.

On day 3, one day after the addition of *C. roenbergensis*, we observed an increased formation of bacterial clumps or aggregates in the water (see Appendix A), which was not observed in the control (Figure 4A,C). While the bacterial population in the water column consisted mainly of clumped *M. adhaerens* cells, in the following days the bacterial clumps decreased progressively in size, correlating with the decline in *C. roenbergensis* abundance.

### 2.2. Surface-Attached Predator Treatment

Overall, *M. adhaerens* dynamics showed a very similar trend to the previously described scenario for planktonic predator treatment. The planktonic *M. adhaerens* subpopulation (Figure 1D) initially increased in cell density until day 2, followed by a sharp drop on day 3, settling on a relatively constant level of low population density until the end of the experiment. After day 3, bacterial biomass in the planktonic habitat was consistently higher in the predation treatment compared to the control. This is supported by results from the GAM analysis indicating a significant positive correlation between *V. anglica* density on the planktonic *M. adhaerens* cell density (Figure 2B, df = 1.000, F = 24.332, *p* < 0.001). The surface-attached subpopulation (Figure 1E) showed an initial increase in density until day 11 in the control treatment and until day 3 in the predation treatment. Maximum density on the surface was significantly higher for the control scenario, as indicated by a significant negative correlation between *V. anglica* density and areal coverage of surface-attached *M. adhaerens* (Figure 3B, df = 3.683, F = 47.441, *p* < 0.001). After the peak, the surface-attached *M. adhaerens* density decreased progressively until the end of the experiment.

Upon addition of the surface-attached predator on day 2 (Figure 1F), *V. anglica* density increased slowly until approximately day 9, after which the predator biomass increased more steeply until day 15, after which it remained at rather high values until the end of the experiment (Figure 1F).

The comparison between control and surface-attached predator treatment indicated that *M. adhaerens* responded to the presence of *V. anglica*, even during the initial phase of low density, via an increased use of the planktonic compartment. This is reflected by the overall higher (lower) density of the planktonic (surface-attached) subpopulation in the predator treatment compared to the respective subpopulation in the control treatment (Figure 1D,E, and Appendix A).

Interestingly, *M. adhaerens* also showed clump formation in the water column after addition of the surface-attached predator (Figure 4B and Appendix A).

In addition to the effects of the predators, the GAM analysis revealed that the density of *M. adhaerens* in the water habitat was significantly negatively correlated to the density of the bacterium in the surface habitat (Figure 2C, df = 1.359, F = 18.512, *p* < 0.001), but not vice versa (Figure 3C, df = 1.827, F = 0.986, *p* = 0.345). Furthermore, planktonic *M. adhaerens* linearly decreased with time (Figure 2D, df = 1.000, F = 77.439, *p* < 0.001), while surface-attached *M. adhaerens* showed a unimodal response over time (Figure 3D, df = 5.469, F = 27.382, *p* < 0.001).

## 3. Discussion

This study investigated how bacterial habitat choice is influenced by the presence of habitat-specific grazers. We observed pronounced differences in biomass accumulation of the chemotaxis mutant of the marine bacterium *M. adhaerens* between the planktonic and surface-attached lifestyles in response to the presence of planktonic vs. surface-attached protist predators. Our observations indicate that *M. adhaerens* escaped to the planktonic habitat in the presence of the surface-attached predator *V. anglica*, whereas it dominantly used the surface-attached habitat in the presence of the planktonic predator. While the molecular basis of the observed bacterial responses and the involved cues are unknown, our results suggest that habitat shifts are key to escape high predation pressure and allow for bacterial persistence. Our results are in line with laboratory experiments by [24] revealing similar escape behavior of four different surface-attached bacteria in the presence of a surface predator—the amoeba *A. castellanii*. They found that even the liquid supernatant of the *A. castellanii* culture induced the bacterial escape response, pointing to the fact that chemical signaling may trigger the bacterial behavior. In particular, the authors showed that at high bacteria-to-amoebae ratios, the presence of the amoebae provoked active bacterial detachment from the surface-attached bacteria into the water column. The observed higher migration of *M. adhaerens* to the planktonic environment in the presence of the surface predator shortly after its addition, when amoeba density was still low, is supportive of such chemical cues.

Apart from predation, differences in nutrient availability between habitats and specifically leaching of organic matter and nutrients from the nutrient-enriched surfaces into the water might also trigger habitat shifts. Moreover, grazing-induced remineralization of the bacterial biomass [25] on the surfaces would further release organic matter and nutrients into the surrounding water and thus stimulate growth of the bacterial subpopulation in the water habitat. However, the stimulation of bacterial growth in the alternative habitat is not obvious from our experimental data, as growth of the planktonic bacteria remained low after day 6, when amoeba density and thus grazing pressure was high.

The defensive response of *M. adhaerens* towards the planktonic predator *C. roenbergensis* seems to be more diverse than for the surface-attached predator, as we observed a morphological response via cell clumping of bacteria in water, in addition to their escape towards surfaces, associated with a drastic decline in the *C. roenbergensis* population. According to this observation, clumping seems to be part of a defensive response from *M. adhaerens* against nanoflagellate grazing in the pelagic environment. Clumping as one possible defense mechanism to avoid predation by heterotrophic nanoflagellates is widely recognized by different studies (e.g., [5]). However, in addition to our observational study, more detailed investigations are necessary to assess the molecular basis of the observed response and how it is related to predator identity and density. The differences (even if small) in bacterial density in the water habitat between the planktonic predator and the control treatment suggest that clumping limited grazing by *C. roenbergensis*. However, the high accumulation of prey biomass on the slides in the presence of the planktonic predator in comparison to the control treatment indicates that the habitat shift was the dominant predator-escape mechanism. While our study did not allow us to assess the role of chemical cues and the molecular basis for the observed responses, other studies (e.g., [26]) suggested that the escape response in bacteria to avoid predation may be mediated by quorum sensing triggered by the increase of bacterial association with particles. Moreover, the production of antiprotozoal substances has been observed for the marine bacterium *Vibrio cholerae* in the presence of the flagellates *C. roenbergensis* and *R. nasuta* [27]. In the latter study, the authors suggested that in addition to less available bacterial prey due to a shift in habitat use (towards the particle surfaces), the antiprotozoal substances caused a drastic reduction in the predator population. This might also explain the strong decline in *C. roenbergensis* in our experiment.

In pelagic habitats, grazing can be mainly by planktonic protozoa or alternatively by surface-attached protozoa, which relate to a different flux of energy and organic matter, as surface-attached protozoa can be also ingested by larger organisms feeding on particles (e.g., suspension feeders like zooplankton, e.g., [28]). In contrast, pelagic protozoa like nanoflagellates can only be ingested by smaller filter feeders (e.g., ciliates), which leads to additional trophic levels in the food chain and hence a less efficient energy and matter flux to higher trophic levels (e.g., [29]). Therefore behavioral defense mechanisms can have consequences for energy flow to higher trophic levels [15].

In a previous study on predator-escape mechanisms of a generalist bacterium (*Pseudomonas putida*) in the presence of either a planktonic (*Paramecium tetraurelia*) or a surface predator (*Acanthamoeba castellanii*), ref. [8] also observed a pronounced habitat shift of the bacteria in response to the respective predation-free habitat. However, ref. [8] only assessed the initial vs. final microbial densities after a rather short experimental duration of only four days. In our study, we applied a much longer duration of 21 days with daily data collection during the first week and every other day for the rest of the experiment. The prolonged experimental time revealed complex microbial dynamics, allowing us to assess the differences in the bacterial response to each of the two habitat-specialized predators. The longer duration of our experiment allowed us to observe clearer differences between control and predation treatments, especially in the surface-predator scenario, where the main differences arose after the first week, probably related to the slower growth of this predator together with a time lag in order to colonize the surfaces. Furthermore, the higher temporal resolution of our experiment allowed us to detect additional bacterial defense mechanisms, such as cell clumping and habitat shift, both of which occurred rapidly in the presence of the planktonic predator. Moreover, ref. [8] used well-plate systems with relatively small volume to assess surface-attached bacteria on the bottom of the well, and this might influence the expression of habitat shifts. In our system, with low-nutrient water and nutrient-enriched surfaces, it seems that nutrients in the water were rapidly taken up and agar-bound nutrients depleted towards the end of the experiment. The decreasing trend of planktonic bacteria over time suggest that bacteria in our batch-culture system were increasingly limited by bottom-up control, especially in the planktonic habitat.

Because of their feeding strategies and dominant habitat use, the two predators *C. roenbergensis* and *V. anglica* can be considered specialist grazers of planktonic bacteria and surface-attached bacteria, respectively. Nonetheless, we also observed the nanoflagellate grazer close to the surfaces (dwelling in the water–surface interphase) when bacteria in the water formed non-ingestible clumps. However, our data do not suggest a strong grazing effect of the nanoflagellate grazer on the surface-associated bacterial subpopulation.

Despite the lack of a chemotactic response towards nutrient patches of the selected *M. adhaerens* chemotaxis mutant (ΔCheA), the observed differences in the response of *M. adhaerens* in the presence of planktonic or surface-attached predators indicate that this strain can still detect the presence of both predators and respond via habitat shift and cell clumping to minimize the impact of protozoan grazing. Whether the defense mechanisms have a chemical basis remains speculative. Future studies could address this aspect and resolve which genes and genetic pathways are involved. It is known that the gene cluster ΔCheA reduces the surface colonization of the marine diatom *Thalassiosira weissflogii* compared to the wild-type strain [30]. The gene knockout may also impact physiological rates, such as the growth rate of bacterial strains. To assess these effects, we measured the growth rates of the chemotaxis knockout mutant (ΔCheA) in comparison to the wild type and also the flagella mutant (ΔFliC) strain in initial experiments (see Appendix A). Under the same experimental conditions as used in the described study (control treatment), both the chemotaxis and the flagella mutant showed a higher growth rate and carrying capacity than the wild-type strain.

While we originally envisioned investigating population dynamics of *M. adhaerens* in the presence of both protist predators (*C. roenbergensis* and *V. anglica*), this was not possible, as in all trials the planktonic predator *C. roenbergensis* went extinct shortly after both predators were added to the system. One main reason for this could be that *C. roenbergensis* reacted sensitively to remainders of the antibiotic, i.e., chloramphenicol, that was used in the solid agar to keep amoeba cultures in axenic conditions prior to starting the experiments. Initial experiments in the absence of chloramphenicol support the view that coexistence between the two predators on a diet of *M. adhaerens* is possible; however, due to the need to use the antibiotic to keep predators axenic and also select for the plasmid fluorescent plasmid encoded on bacterial plasmids, this setup was not used for the main experiment.

## 4. Materials and Methods

### 4.1. Organisms and Culture Media

In our laboratory experiments, we used the Gram-negative gammaproteobacterium *M. adhaerens* (HP 15). The strain was originally isolated from the Wadden Sea and firstly described by [31] as a marine particle-associated bacterium and had its genome sequenced by [32]. For the experiments, we selected the knockout chemotaxis mutant (ΔCheA), tagged with a plasmid coding for a red fluorescence protein (DsRed) [33,34]. The chemotaxis mutant shows both planktonic and surface-associated lifestyles, which allows studying the behavioral response of *M. adhaerens* in presence of either the planktonic or the surface-associated protist predator. The knockout gene of the chemotaxis mutant has been shown to reduce colonization rates on cells of the diatom *Thalassiosira weissflogii* with respect to the wild-type strain [30], which indicates that colonization is less driven by resource availability. Furthermore, initial experiments showed that the chemotaxis mutant (from here on called *M. adhaerens*) exhibited the highest growth rate compared to the wild type or the flagella mutant (see Appendix A). Both of these observations made it more likely to observe pronounced differences between the treatments and to observe habitat shifts primarily driven by predator abundance and not resource availability.

*M. adhaerens* was stored in cryos at −80 °C, keeping axenic conditions. For bacterial growth, we used marine broth (MB) medium with ampicillin 30 µg/mL to select for the fluorescence protein-tagged bacteria, due to the presence of ampicillin and chloramphenicol resistance encoded in the plasmid. The bacterial growth temperature was 37 °C and the tubes were incubated overnight. During the exponential growth phase, 2 mL of the culture were centrifuged at 10,000 rpm for 1 min and the pellet was washed with artificial seawater (ASW; ref. [35]). After resuspending the pellet with ASW, we repeated the washing process 3 times in order to get rid of any nutrients from the growth medium (MB).

For protozoan grazers, we used (i) the nanoflagellate *Cafeteria roenbergensis* [36] as planktonic predator and (ii) the amoeba *Vannella anglica* [37] as surface-associated predator. For *C. roenbergensis* cultures, we used ASW plus soil extract and a wheat grain previously autoclaved. The amoeba *V. anglica* was kept on agar plates using MY75S medium containing malt, yeast, and 75% artificial sea water (ASW; ref. [35]) plus bacteriological agar (15 g L^−1^). Both organisms were treated with a mixture of antibiotics over several generations in order to reduce the background bacterial density. Cultures for the experiments were achieved by diluting the *C. roenbergensis* culture 20 times. Thereafter, a mixture of antibiotics at 30 µg/mL was added to the liquid medium and the whole process was repeated over 5 generations. To increase flagellate density, prior to the experiment, we fed the culture with heat-killed *M. adhaerens*. For the amoeba *V. anglica*, after autoclaving and cooling, a mix of antibiotics was added to the agar solution before pouring the agar solution into the plates.

### 4.2. Model System and Grazing Experiments

We developed a laboratory experimental system comprising the two pelagic habitats: water (planktonic) and solid-particle surfaces (surface-associated). We used a semicontinuous batch-culture system with three replicates per treatment of 800 mL glass beakers filled with 200 mL of ASW (water) containing 4 microscope glass slides (particle surface), exposed vertically in the water. One side of the glass slides was coated with solid agar solution. The beakers were closed in order to avoid any external contamination. To keep the temperature constant throughout the experiment, we used an incubator set at 20 °C, representative of surface-water summer temperatures of temperate regions. To mimic conditions of the pelagic environment, we used a shaker to gently mix the batch cultures at low intensity (approx. 0.25 s^−1^ of shear rate), trying to mimic the average turbidity regime of ocean surface waters.

The liquid medium consisted of ASW with carbon (C) and nutrient (N, P) concentrations mimicking nitrogen limitation based on the Redfield ratio [38], representative of oligotrophic marine environments. We used 20 µM glucose, 2 µM ammonium, and 0.2 µM phosphate as C, N, and P source, respectively. Micronutrients were delivered with f/2 medium [39], which was diluted 1000× to obtain low trace metal and vitamin concentrations similar to oligotrophic marine environments. The solid surfaces of the glass slides were covered with a solution of ASW with 5% bacteriological agar and two orders of magnitude (100×) more C, N, and P than in the water habitat, simulating nutrient enriched organic particles. C, N, and P content were 2000 µM (glucose), 200 µM (ammonium), and 20 µM (phosphate), respectively. After autoclaving, 1.2 mL of the enriched agar solution was poured onto the glass microscope slides and spread homogeneously on one side of the slide, covering an area of 3.5 × 2.5 cm. The slides were vertically introduced into the beakers with the agar solution-covered side facing upward at an angle of approx. 45°. This homogeneous distribution on one side allowed us to better assess bacterial and predator density via microscopy. In both habitats, ammonium (NH_4_^+^) was the limiting nutrient, allowing us to follow nutrient dynamics throughout the experiment by measuring ammonium (NH_4_^+^), nitrite (NO_2_^−^), and nitrate (NO_3_^−^) (see Appendix A). Ammonium leaching time from the enriched glass surfaces into the water was calculated using Fick’s first law, which suggests that ammonium concentrations in the water and particle surfaces reach equilibrium in less than one hour (See Appendix A).

To evaluate the escape ability of the generalist bacterium *M. adhaerens*, we followed its abundance in both habitats—in the water and on the slide surfaces—with three experimental treatments: (i) control treatment with only *M. adhaerens*, i.e., absence of both predators, (ii) planktonic predator treatment with *M. adhaerens* and *C. roenbergensis*, and (iii) surface-attached predator treatment with *M. adhaerens* and *V. anglica*. Unfortunately, it was not possible to investigate *M. adhaerens* dynamics in the presence of both predators, since the nanoflagellate (*C. roenbergensis*) quickly went extinct in the presence of the amoeba *V. anglica*. Presumably, this was due to the (albeit low) concentrations of chloramphenicol released from the agar block used for amoeba inoculation (see also Discussion).

All three treatments were triplicated with initial bacterial cell density of 10^4^ cells mL^−1^. The microbial cells were taken from the same stock culture and at the same growth phase to ensure the same cell physiological status between treatments. Bacterial growth (in absence of grazers) was observed to reach carrying capacity after two days, as measured in initial experiments (Appendix A). Accordingly, for the predator treatments, the respective predator was added on the second day of the experiment, when bacterial abundance had reached >10^6^ cells mL^−1^ (Figure 1). The initial density of the planktonic predator *C. roenbergensis* was 10^4^ cells mL^−1^, the surface-attached predator *V. anglica* was added with 4 × 10^4^ cells on 4 mm^2^ agar blocks (after gently centrifuging the agar blocks to detach the loosely attached *V. anglica*), and they were added to the cultures.

The experiments lasted 21 days to give the predators enough time for a numeric response. The amoeba *V. anglica* has a low growth rate compared to the nanoflagellate *C. roenbergensis*. Sampling was carried out daily during the first week and every second day during the second and third weeks. Twelve milliliters of water was taken out from each batch culture under sterile conditions in a clean bench. From these, after prefiltration through 0.22 µm syringe filters, 5 mL was used to measure nutrients with a FIASTAR analyzer after 1:2 dilution using nutrient-free ASW. From this, 5 mL was used to measure the density of the nanoflagellate, *C. roenbergensis* in the planktonic habitat, after fixing with glutaraldehyde 2% (fin. conc.) and storage in the fridge at 4 °C for a maximum of 4 weeks prior to filtration. Finally, 1 mL was used to determine bacterial cell density after filtration on 0.2 µm filters to quantify the fluorescence signal of the protein (DSRed) encoded in a plasmid within *M. adhaerens* with an epifluorescence microscope (Leica DMRB). The excitation and emission maxima of the fluorescence protein DSRed are 554 and 586 nm, respectively. The 12 mL sampled was replaced by adding new medium on every sampling day to keep the 200 mL constant along the experiment. Note that the sampling timing differed between the first week and the rest of the experiment, and thus the dilution rate was not constant over time. However, given the small volume taken for sampling, the corresponding dilution rate was rather low, ranging from 0.06/day in the first week to 0.03/day in the second and third weeks. We therefore assume that this did not substantially affect the observed population dynamics (see Figure 1). To assess bacterial and protist densities on the solid surfaces, at each sampling, life measurement was taken using (i) an inverted microscope for direct investigations of protists and (ii) an epifluorescence microscope assessing the density of *M adhaerens*, making use of the fluorescence DSRed protein encoded in the bacterium *M. adherens*. Differently to our expectation, the nanoflagellate planktonic predator has also been found loosely associated with the agar surfaces, but feeding mainly on plankton and not on attached bacteria [21].

We always took 9 pictures along each slide, corresponding to three lines along the slide’s vertical axes (left, middle, right) and three points per vertical line (upper, medium, and lower). Additional picture capture did not substantially affect the biomass output; therefore, 9 pictures were established to measure the biomass on the slides. ImageJ was used to calculate the fraction [%] of the surface-area covered by surface-attached bacteria (see Appendix A) and to approximate the density of bacteria attached to the agar surfaces. To assess the bacterial density in the water, we counted 20 microscope squares randomly distributed across each filter. Predator density on the surface slides, both planktonic (see above comment) and surface-attached, were quantified by counting 40 consecutive randomly distributed squares using the inverse microscope. For quantification of predator density in the water, 5 mL of the sampling volume was fixed with 2% glutaraldehyde (fin. conc.), filtered on 0.8 µm white filters, stained with 2.5 µg/mL DAPI (fin. conc.), and counted using the epifluorescence microscope by counting 40 randomly distributed squares.

For additional experimental details concerning our batch-culture system, see Appendix A.

### 4.3. Statistical Analysis

To investigate factors determining the population density of *M. adhaerens* in both habitats, we used a generalized additive model (GAM). GAM is a method to estimate flexible regression functions between explanatory and response variables [40]. We constructed two GAM models to explain either the density of planktonic or surface-attached *M. adhaerens*. For this, we analyzed all data together (control, planktonic predator treatment, and surface predator treatment). In the first model, the density of *M. adhaerens* in the water habitat was set as the response variable. The density of the predators *C. roenbergensis*, *V. anglica*, and *M. adhaerens* in the surface-attached habitat (on slides) and time were selected as the explanatory variables. Thereby the covariate function depending on the explanatory variable “time” indicated the changes in population density influenced by factors other than predation. In our analysis, we focused on the predator effects (via *C. roenbergensis* or *V. anglica*) and bacterial density in the alternative habitat (i.e., in water or on surfaces) by further analyzing the corresponding functional terms indicated by the GAM analysis. We performed the same analysis with respect to the density of *M. adhaerens* on the slides as the response variable in the second model. In this case, the density of *C. roenbergensis*, *V. anglica* and *M. adhaerens* in the water and time were selected as the explanatory variables. Please note that there was one control treatment that was used for both models, representing the case of zero predator density.

The data analysis was performed on the sampling points, starting with day 2 of the experiment on which the predators had been added. Prior to analysis, the data were divided by the corresponding standard deviation to exclude the effects of differences of variances among the explanatory variables. For the GAM analysis a gamma error distribution with the log-link function was used to account for the fact that the response variables were positive only. The GAM analysis was performed using the mgcv package (V1.8-31) in R (V3.6.2R Core Team (2021)).

For additional experimental details concerning the statistical GAM analysis, see Appendix A.

## 5. Conclusions

We demonstrate the presence of contrasting defensive mechanisms in the chemotaxis mutant *M. adhaerens* (ΔCheA) towards planktonic or surface attached predators in the two main pelagic habitats: water and particle-surfaces. We found multiple plastic responses (i.e., escape into the predator-free habitat and cell clumping) against predator grazing, furthermore bacterial defense-responses differed in response to the planktonic or surface-attached predator. The presence of active predator-escape mechanisms affects species interactions and community composition with potential consequences for the flux of energy and matter through the microbial loop. Our study highlights, that taking microbial interactions into account might be crucial to assess how changing abiotic conditions via nutrient availability and particle concentration will impact energy transport and remineralization in aquatic systems. Thereby the underlying genetic mechanisms of *M. adhaerens’s* defense responses are yet unknown and future efforts should shed light on the molecular basis and possible eco-evolutionary processes that regulate predator-prey interactions in microbial communities.

## Figures and Tables

**Figure 1 ijms-23-10082-f001:**
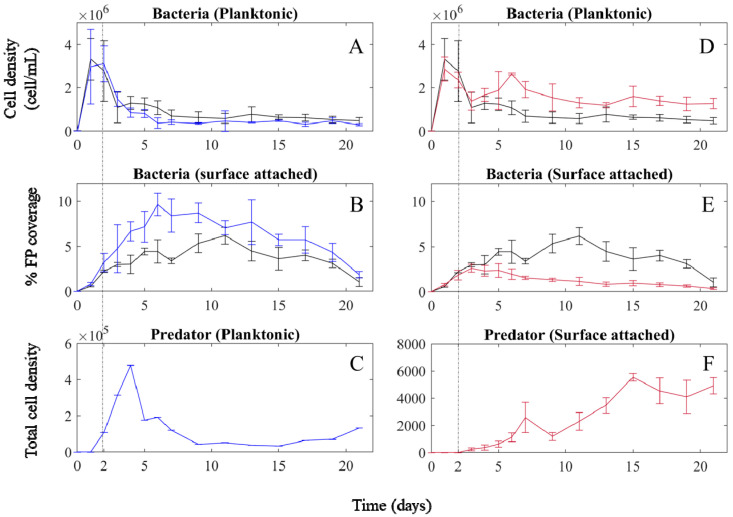
Predator–prey dynamics comparing bacteria population dynamics in the water (**A**,**D**) and attached to surfaces (**B**,**E**) in presence of (**A**–**C**) the planktonic predator (blue), or (**D**–**F**) the surface-attached predator (red) with the corresponding dynamics observed in the control treatment (black line). (**C**,**F**) show the total cell density (light blue solid line) of the planktonic *C. roenbergensis* ((**C**), blue) and the total cell density per total surface area (red solid line) of the surface-attached *V. anglica* ((**F**), red, respectively. The vertical dashed black line marks the addition of the respective predator on day 2 of the experiment.

**Figure 2 ijms-23-10082-f002:**
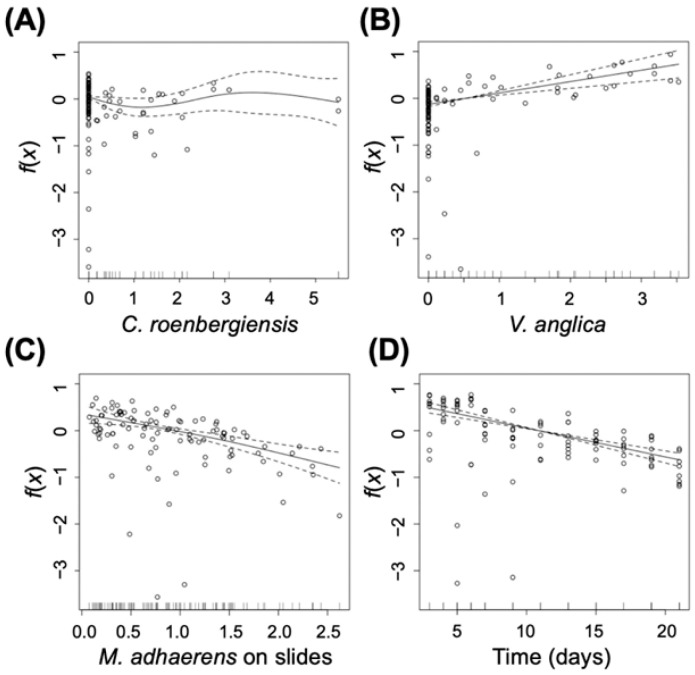
Smoothed fits of covariates modeling the scaled density of planktonic *M. adhaerens* for the scaled densities of (**A**) *C. roenbergensis*, (**B**) *V. anglica*, (**C**) surface-associated *M. adhaerens*, and (**D**) time. The *y*-axis represents the spline function. Circles are the data from the experiment. Solid lines represent fitting curves. Dashed lines indicate the 95% confidence intervals.

**Figure 3 ijms-23-10082-f003:**
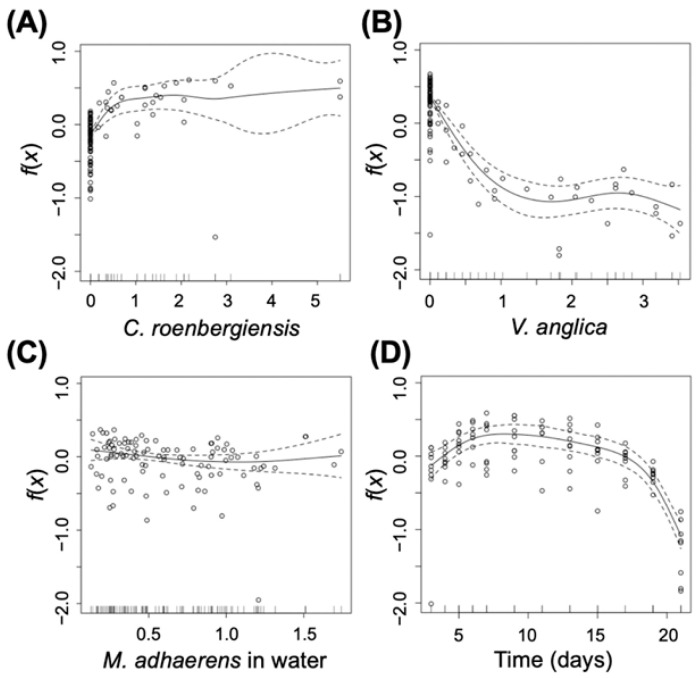
Smoothed fits of covariates modeling the scaled density of surface-attached *M. adhaerens* for the scaled densities of (**A**) *C. roenbergensis*, (**B**) *V. anglica*, (**C**) planktonic *M. adhaerens*, and (**D**) time. The *y*-axis represents the spline function. Circles are the data from the experiment. Solid lines represent fitting curves. Dashed lines indicate the 95% confidence intervals.

**Figure 4 ijms-23-10082-f004:**
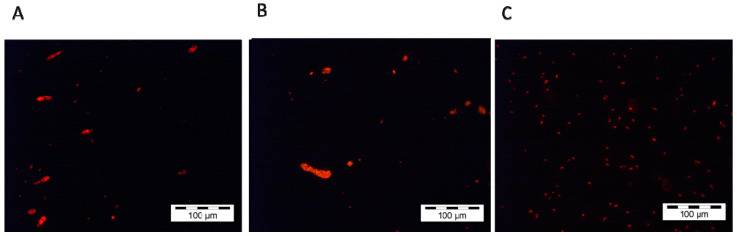
Documentation of clump formation in the water habitat on day 3 in presence of (**A**) *C. roenbergensis*, (**B**) *V. anglica*, and (**C**) absence of clumps in the control treatment. Pictures were taken with a Leica epifluorescence microscope at 40× magnification. The red depicts the fluorescence of the DSRed protein.

## Data Availability

Not applicable.

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
