# Peer review of "Differing Escape Responses of the Marine Bacterium Marinobacter adhaerens in the Presence of Planktonic vs. Surface-Associated Protist Grazers"

_ijms, 2022, doi:10.3390/ijms231710082_

Round 1
Reviewer 1 Report
I appreciate the study undertaken by Villalba et al. entitled “Differing escape-responses of the marine bacterium Marinobacter adhaerens in the presence of planktonic vs. surface-associated protist grazers”. I would request the authors to make some clarifications regarding the following concerns that I have for this study:
1. I liked the concept of specialist and generalist in the paper. It will be helpful for the readers if the concept of specialist and generalist can be introduced in the introduction section. Also, if the terms can be defined.
2. Figure 4: Please mention the scale bars of the fields under the epifluorescence microscope.
3. I must say that the discussions are far-fetched and should be rephrased or the authors should be more cautious while claiming the statements mentioned in lines# 463-464 – what experimental evidence do the authors have regarding flux of energy and matter through the microbial loop?
4. The authors should explain why chemotaxis mutant Marinobacter was used for this study.
5. Minor concerns:
Lines# 302: Gram and not gram
Lines# 417: established
Lines#454: reformat the reference
Author Response
I appreciate the study undertaken by Villalba et al. entitled “Differing escape-responses of the marine bacterium Marinobacter adhaerens in the presence of planktonic vs. surface-associated protist grazers”.
Many thanks for your motivating comments.
I would request the authors to make some clarifications regarding the following concerns that I have for this study:
- I liked the concept of specialist and generalist in the paper. It will be helpful for the readers if the concept of specialist and generalist can be introduced in the introduction section. Also, if the terms can be defined.
We added some sentences to clarify the definition of specialist versus generalist used in this manuscript. (see page 2, L 61-76).
- Figure 4: Please mention the scale bars of the fields under the epifluorescence microscope.
We have now added scale bars to the respective images of figure 4. (See page 6).
- I must say that the discussions are far-fetched and should be rephrased or the authors should be more cautious while claiming the statements mentioned in lines# 463-464 – what experimental evidence do the authors have regarding flux of energy and matter through the microbial loop?
We are sorry that the impression of overstatements was raised. This was not our intention. The statements on the relevance of habitat shifts for energy flow are based on indications from a previous theoretical study (Villalba et al. 2022). We tried to make this clearer and rephrased certain paragraphs in the introduction as well as the discussion accordingly, see page 2 and 3 (introduction) and page 7 and 8 (discussion).
- The authors should explain why chemotaxis mutant Marinobacter was used for this
We were looking for a bacterium with the ability to shift between the surface and the planktonic habitat. Our choice was based on previous results on the performance of Marinobacter, showing that the chemotaxis knockout mutant reduced the colonization on the diatom Thalassiosira weissflogii with respect to the Wild Type (WT) (Sonnenschein et al. 2012), thus, the habitat migration based on substrate availability is limited and we were able to mainly focus on predation based migration. Furthermore, in our pre-experiments we show that this chemotaxis mutant strain was better grower in both habitats (water and particles) than the WT under the experimental conditions.
Both reasons made the chemotaxis mutant an ideal candidate to study habitat shifts in presence of planktonic or surface-associated predators. We extended and reshaped the explanations in the Method section accordingly (see page 9, L 350-357).
- Minor concerns:
Lines# 302: Gram and not gram
Has been corrected. (see page 9, L341)
Lines# 417: established
Has been corrected. (see page 11, L460)
Lines#454: reformat the reference
Has been corrected. (see page 12, L498)
Reviewer 2 Report
Villaba and colleagues investigated the escape-response of the M. adhaerens in the presence of either planktonic or surface-associated protist predators. They concluded that the cell density of M. adhaerens increased in water in the presence of surface-associated protist predator (V. anglica). In contrast, the cell density of M. adhaerens increased in the solid surface in the presence of planktonic protist predator (C. roeneorgensis).
The results are interesting but preliminary. However, the exact cause and mechanism were not profoundly investigated. Therefore, some of the interpretations lacked evidence and should be cautious. For example, the cell clumping of bacteria in water in addition to their escape towards the surfaces.
There are several typos in the appendix should be carefully revised.
1. In appendix E, ammonium instead of ammonium.
2. In appendix G, Use of ImageJ for calculating bacterial “number” on particle surfaces. I wonder if the number means density ?
Author Response
Villaba and colleagues investigated the escape-response of the M. adhaerens in the presence of either planktonic or surface-associated protist predators. They concluded that the cell density of M. adhaerens increased in water in the presence of surface-associated protist predator (V. anglica). In contrast, the cell density of M. adhaerens increased in the solid surface in the presence of planktonic protist predator (C. roeneorgensis).
The results are interesting but preliminary.
Thank you very much for your overall positive statement.
However, the exact cause and mechanism were not profoundly investigated. Therefore, some of the interpretations lacked evidence and should be cautious. For example, the cell clumping of bacteria in water in addition to their escape towards the surfaces
It was not the aim of this study to study the exact causes and underlying mechanisms as we didn’t use molecular techniques such as transcriptomes. Instead, we wanted to follow changes in bacterial behavior (habitat use) in presence of different habitat specialist grazers. For this we investigated the biomass development in the two compartments over time. We revised some passages in the discussion to avoid any overstatement of our results and more clearly address the limitations of our study in assessing the underlying mechanisms (page 7-8).
There are several typos in the appendix should be carefully revised.
- In appendix E, ammonium instead of ammonium.
Has been corrected.
- In appendix G, Use of ImageJ for calculating bacterial “number” on particle surfaces. I wonder if the number means density ?
We rephrased the respective sentence in the Methods Section to make clear that we use this value as a proxy for bacterial density in the agar surfaces. (see page 11, L 460-463).
Round 2
Reviewer 1 Report
The manuscript was revised satisfactorily.